# Advanced Sperm Selection Strategies as a Treatment for Infertile Couples: A Systematic Review

**DOI:** 10.3390/ijms232213859

**Published:** 2022-11-10

**Authors:** Jordi Ribas-Maynou, Isabel Barranco, Maria Sorolla-Segura, Marc Llavanera, Ariadna Delgado-Bermúdez, Marc Yeste

**Affiliations:** 1Unit of Cell Biology, Department of Biology, Faculty of Sciences, University of Girona, 17003 Girona, Spain; 2Biotechnology of Animal and Human Reproduction (TechnoSperm), Institute of Food and Agricultural Technology, University of Girona, 17003 Girona, Spain; 3Department of Veterinary Medical Sciences, University of Bologna, Ozzano dell’Emilia, 40126 Bologna, Italy; 4Catalan Institution for Research and Advanced Studies (ICREA), 08010 Barcelona, Spain

**Keywords:** fertility, ICSI, infertility, IVF, sperm selection techniques, sperm quality

## Abstract

Assisted reproductive technology (ART) is an essential tool to overcome infertility, and is a worldwide disease that affects millions of couples at reproductive age. Sperm selection is a crucial step in ART treatment, as it ensures the use of the highest quality sperm for fertilization, thus increasing the chances of a positive outcome. In recent years, advanced sperm selection strategies for ART have been developed with the aim of mimicking the physiological sperm selection that occurs in the female genital tract. This systematic review sought to evaluate whether advanced sperm selection techniques could improve ART outcomes and sperm quality/functionality parameters compared to traditional sperm selection methods (swim-up or density gradients) in infertile couples. According to preferred reporting items for systematic reviews and meta-analyses (PRISMA guidelines), the inclusion and exclusion criteria were defined in a PICOS (population, intervention, comparator, outcome, study) table. A systematic search of the available literature published in MEDLINE-PubMed until December 2021 was subsequently conducted. Although 4237 articles were recorded after an initial search, only 47 studies were finally included. Most reports (30/47; 63.8%) revealed an improvement in ART outcomes after conducting advanced vs. traditional sperm selection methods. Among those that also assessed sperm quality/functionality parameters (12/47), there was a consensus (10/12; 83.3%) about the beneficial effect of advanced sperm selection methods on these variables. In conclusion, the application of advanced sperm selection methods improves ART outcomes. In spite of this, as no differences in the reproductive efficiency between advanced methods has been reported, none can be pointed out as a gold standard to be conducted routinely. Further research addressing whether the efficiency of each method relies on the etiology of infertility is warranted.

## 1. Introduction

Alarming data indicate that human fertility is constantly decreasing, which leads to the performance of around 1435 assisted reproductive technology (ART) cycles per million of habitants in Europe every year [1]. Infertility is considered as a disease by the World Health Organization, with an estimated incidence of 8–12% couples at reproductive age worldwide [2]. A male factor is involved in about half of infertility cases, being either the main affectation or a cofactor affecting couple’s fertility. Male factors are related to physiological alterations, such as hypogonadism, erectile dysfunction or retrograde ejaculation, and impaired sperm quality [3,4]. When infertility is due to low sperm quality, ART treatments, such as in vitro fertilization (IVF) or intracytoplasmic sperm injection (ICSI), become an alternative to achieve fertilization [4]. While the understanding of ART techniques has increased and great strides to develop novel technologies maximizing their success have been performed, recent data evidence that there are still significant shortcomings to reach satisfactory pregnancy and live birth rates [5].

In the last two decades, research in ART has concentrated on better identifying which factors underlie infertility and, thus, how their handling may increase the chances of successful pregnancy. At present, sperm quality is understood to be one of the key features driving a decrease in both fertilization rates and the proportion of embryos successfully developing to blastocyst stage [6,7]. In addition, mounting evidence suggests that an impairment of sperm DNA integrity causes a reduction of ART outcomes [8]. For this reason, the selection of the most competent sperm cells—i.e., those with the highest fertilizing ability—is crucial to improve both laboratory and clinical outcomes following ART. In this regard, it is worth noticing that sperm selection based on physiological or molecular features has gained much interest from researchers [9,10]. In in vivo fertilization, sperm are subject to natural selection during their journey alongside the female reproductive tract, involving dynamic and morphological features [11]. However, this selection process can be bypassed through ART (mainly ICSI) and, as a result, sperm with alterations may be used to fertilize an oocyte [10]; because of that, sperm selection methods mimicking the female genital tract are the focus of research [12]. Whilst traditional sperm selection techniques, mainly based on sedimentation or migration (swim-up and density gradient centrifugation), are useful to select motile, morphologically normal sperm, there is still room for improvement regarding their efficiency. For this reason, novel selection methods based on other sperm features, such as ultrastructure, surface cell proteins, or DNA integrity, have been developed in the last years [13,14]. These approaches increase the likelihood of selecting structurally intact, viable, and mature sperm with an intact DNA prior to ART, thus enhancing fertilization, embryo development, and pregnancy rates.

Herein, critical literature assessing the effects of conventional and advanced sperm selection methods was systemically reviewed, with the purpose of elucidating whether the latter can be used to improve sperm quality/functionality parameters and/or ART outcomes.

## 2. Material and Methods

Preferred reporting items for systematic reviews and meta-analyses (PRISMA) guidelines [15] were followed to conduct the systematic review. The search protocol was registered in the PROSPERO registry (http://www.crd.york.ac.uk/PROSPERO (accessed on 10 November 2021)) with the number PROSPERO 2021 ID: CRD42021248949.

### 2.1. Data Sources and Search Strategy

The MEDLINE-Pubmed database (http://www.ncbi.nlm.nih.gov/pubmed (accessed on 13 December 2021)) was utilized to conduct a systematic search of the available literature, which included research studies published until 13 December 2021. Inclusion and exclusion criteria for the selected studies were defined prior to the search in a *Population, Intervention, Comparator, Outcome, Study* (PICOS) table (Table 1). Based on this table, a list of keywords was set and used for the definition of the search strategy as follows: (*infertile OR infertility OR sperm selection OR sperm selection methods OR sperm selection techniques) AND (MSOME OR motile sperm organelle morphology examination OR birefringence OR polarized light microscope OR Raman OR microfluidics OR IMSI OR hyaluronic OR ICSI OR MACS OR magnetic activated cell sorting OR swim-up OR density gradients OR Zeta-potential OR electrophoresis OR Annexin V) AND (Assisted reproduction OR AI OR artificial insemination OR insemination OR IVF OR* in vitro *fertilization OR ICSI OR intracytoplasmic sperm injection OR intrauterine insemination OR IUI) AND (sperm quality OR sperm function OR motility OR DNA damage OR DNA fragmentation OR oxidative stress OR free radicals OR viability) AND (embryo OR blastocyst OR zygote OR fertility OR pregnancy OR implantation OR live birth OR fertilization)*. In addition, the filter ‘*Species (Humans)*’ was applied to comply with the inclusion criteria:

### 2.2. Study Eligibility

Articles meeting the inclusion criteria previously defined in the PICOS table (Table 1) were considered in this systematic review. The main criteria were: (i) studies conducted in humans; (ii) articles using an advanced sperm selection technique; (iii) studies comparing advanced vs. traditional sperm selection techniques (e.g., swim-up, density gradients); (iv) studies analyzing sperm quality/functionality parameters before and after applying an advanced sperm selection method; (v) studies assessing the potential influence of advanced sperm selection methods on assisted reproduction outcomes. The main exclusion criteria were: (i) studies conducted in species other than humans; (ii) studies applying an advanced sperm selection method without comparison to traditional sperm selection methods (such as swim-up or density gradients); (iii) studies that did not evaluate sperm quality/functionality parameters and/or fertility outcomes after ART. Research articles, observational studies, cross-sectional studies, comparative studies, and longitudinal studies were eligible, whereas meta-analyses, narrative and systematic reviews, letters, commentary articles, and case reports were declared as non-eligible.

### 2.3. Study Selection Procedure

The study selection procedure was performed following the flowchart displayed in Figure 1. After conducting the search at MEDLINE-PubMed, the article list was downloaded as a .txt file using a standardized data extraction (PMID format separated by tabs) with the following information: Pubmed ID, publication date, authors, title, keywords, document type, journal, ISSN, DOI, and abstract. Then, an Excel file including this information was generated. For eligibility, all information included in the Excel file was examined by two researchers (M.S. and J.R.-M.), and any discrepancy was re-evaluated by a third author (I.B.). First, articles declared as non-eligible and/or not written in English were excluded. Second, studies were selected on the basis of their title and abstract, and those that did not meet the eligibility criteria stated in the PICOS table were excluded. Finally, the full text of selected articles was downloaded and read carefully, and the content was analyzed. This led to the final list of articles included in the systematic review.

### 2.4. Additional Article Quality Screening

An additional step for article quality analysis was performed following NHLBI-NIH guidelines. Specifically, the quality assessment tool for case-control studies http://www.nhlbi.nih.gov/health-topics/study-quality-assessment-tools (accessed on 20 December 2021), which includes 12 YES/NO questions assessing potential weaknesses that may compromise the quality of a study, was applied. By adding up the answers from the questions, a score value between 0 and 12 was obtained. Studies with a score value < 5 were classified as with “poor quality” and, therefore, excluded from the systematic review.

### 2.5. Data Extraction for Systematic Review and Statistics

Once articles were selected, they were analyzed to extract the following data: (1) reference, (2) aim of the study, (3) advanced sperm selection technique applied, (4) sample size, (5) female/male inclusion/exclusion factors, (6) fertility/sperm quality and functionality parameters assessed, (7) main results, and (8) conclusions. In addition, an additional row indicating whether the application of the advanced sperm selection method improved at least one fertility/sperm quality or functionality variable was included for each study.

### 2.6. Statistical Analysis

The Chi-square test was run to determine whether advanced sperm selection methods improve sperm quality/functionality parameters and/or ART outcomes. The statistical significance level was set at 95% of the confidence interval (*p* ≤ 0.05).

## 3. Results

### 3.1. Identification and Selection of Studies

Figure 1 shows how articles were selected, referring to inclusion/exclusion reasons. A total of 4237 articles were recorded after the initial search, and after an initial screening for article type and language, and a secondary screening for title and abstract, a total of 127 articles were selected for full-text assessment. Thereafter, 80 articles were excluded based on the criteria established in the PICOS table (Table 1), and one article was excluded based on quality assessment according to the NHLBI-NIH guidelines (Appendix A). Therefore, a total of 47 studies were declared eligible for qualitative analysis (Figure 1).

### 3.2. Systematic Review: Qualitative Analysis

The 47 studies included in the systematic review were analyzed to extract the key data as described in the Materials and Methods section and summarized in Table 2 and Table 3. According to the sperm selection technique, 21 articles (21/47; 44.7%) used intra-cytoplasmic morphologically selected sperm injection (IMSI), also named motile sperm organelle morphology examination (MSOME), ten (10/47; 21.3%) performed hyaluronic acid selection (physiological ICSI, PICSI), seven (7/47, 14.9%) conducted magnetic-activated cell sorting (MACS) for Annexin V, five (5/47, 10.6%) utilized microfluidic devices, two (2/47, 4.3%) used Zeta-potential, one study (1/47; 2.1%) conducted birefringence, and one study (1/47, 2.1%) carried out laser beam selection.

### 3.3. Influence of Sperm Selection Methods on Sperm Quality and ART Outcomes

Regarding ART outcomes, 30 out of the 47 studies (63.8%) revealed an improvement in fertility results after applying advanced sperm selection methods. Specifically, and sorted by sperm selection method, ART outcomes were found to be improved in 12 of the 21 articles (57.1%) using IMSI or MSOME; in seven of the 10 articles (70%) conducting PICSI; in six of the seven articles (85.7%) performing MACS; in two of the five articles (40%) utilizing a microfluidic device; and in one of the two articles (50%) using Zeta-potential. Studies assessing birefringence and laser beam were also able to boost sperm fertilizing ability. The percentage of studies finding an improvement of ART outcomes, however, did not differ between sperm selection techniques (*p* = 0.585).

The studies reporting ART outcomes also determined the potential effects on sperm quality/functionality parameters that were extracted so as to address whether an improvement on ART outcomes was associated with an increase in sperm quality/functionality variables. Twelve studies assessed sperm quality/functionality parameters before and after semen samples were subject to sperm selection. Among them, ten articles (83.3%) showed a beneficial effect of advanced sperm selection methods on sperm quality/functionality parameters compared to the control. Nine of these ten studies also found an improvement in ART outcomes (90%), thus showing that such a beneficial effect was concomitant with an increased sperm quality.

## 4. Discussion

The present study systematically reviewed the available literature to address whether using advanced sperm selection techniques before ART treatment (mainly ICSI) improves ART outcomes. After applying the defined inclusion/exclusion criteria and assessing the quality of studies, 47 articles were chosen. The advanced sperm selection methods in most of these studies were IMSI, PICSI, or MACS; in contrast, laser, birefringence, zeta-potential, and microfluidics were less investigated.

### 4.1. Using Advanced Sperm Selection Techniques Improves ART Outcomes

Most studies included in this systematic review (30/47) showed an improvement of ART outcomes when semen samples were subject to advanced sperm selection techniques, in comparison to conventional sperm selection methods. However, no advanced sperm selection technique was found to be better than another, as the use of the same method in a similar cohort brought inconsistent outcomes.

IMSI, also known as MSOME, selects sperm without vacuoles in the cytoplasm under a microscope with high magnification (over 6000×) [63]. Using this method, most of the studies compiled herein (57.2%) revealed an improvement in ART outcomes [16,18,20,22,25,26,28,30,33,35,36]. Other articles (42.8%), however, did not find differences between advanced and traditional sperm selection methods regarding ART outcomes [19,21,23,24,29,31,32,34]. Differences in laboratory/clinical procedures or in exclusion/inclusion factors, such as patient selection criteria (age, female factors or male factors), could be behind these inconsistent results. For instance, all studies that limited the female age as an inclusion criterion observed greater fertilization, blastocyst, implantation, pregnancy and live birth rates, and higher embryo quality [16,18,25,26,27,28,30,33,35,36]. Moreover, certain clinical characteristics could also condition the success of IMSI in improving ART outcomes. Indeed, Oliveira et al. [32] and Gatimel et al. [24] did not find statistically significant differences in pregnancy and implantation rates between IMSI and conventional ICSI in patients with recurrent implantation failure. Similarly, the study by Boediono et al. [21] included females with severe endometriosis and observed no improvement in clinical pregnancy rates. Not only do these studies suggest that the success of IMSI relies on the patient, but they also support that further research should include a larger sample size to establish in which clinical conditions IMSI could bring the most benefit.

As far as sperm selection by PICSI is concerned, most of the included studies (70%) reported an improvement on ART outcomes compared to conventional selection methods [38,39,40,43,44,45,46]. Selection through PICSI relies on a specific receptor present in mature sperm, which allows them to bind hyaluronic acid, a component surrounding cumulus cells. This technique thus mimics the natural selection of sperm upon interaction with oocyte vestments [64]. Amongst all studies, those that investigated the male factor of infertility showed a statistically significant improvement on ART outcomes [38,39,40]. The effects of PICSI were more evident when the age of the females was controlled as an inclusion factor. Three studies that limited female age as an inclusion criterion, studying females up to 38–40 years old, proved that PICSI showed higher ART outcomes than conventional ICSI [43,45,46]. Because female age is well known to be a factor that impacts negatively on ART outcomes, mainly due to an increase of oocyte aneuploidies [65], the assessment of younger females could help reduce the bias detected. In fact, all studies that did not find an improvement of ART outcomes after PICSI selection included females of up to 42–43 years old [37,41,42]. Other studies conducted in animal models, such as the pig, found an increase of embryo euploidy after conducting ICSI with hyaluronic acid-selected sperm cells [66]. Hence, while a high proportion of studies supports the use of PICSI to select sperm, this technique is more suitable when the oocyte comes from a young female.

With regard to MACS, this method is based on the conjugation of Annexin V to magnetic microspheres, which are exposed to a magnetic field in an affinity column that allows an effective separation of apoptotic from non-apoptotic sperm [67]. Most research works (85.7%) focused on this technique found an improvement on ART outcomes compared to conventional sperm selection methods. In four studies, female exclusion factors were applied (advanced female age, hormonal levels, number of oocytes after ovarian stimulation, or other physiological alterations) and an improvement in blastocyst quality, implantation, pregnancy rates, and a decrease in miscarriage rates were observed [47,50,52,53]. In two studies, male factors based on sperm quality/functionality parameters were also controlled as inclusion factors, and an improvement of MACS vs. conventional ICSI regarding the quality of blastocysts and a lower rate of miscarriages were reported [50,52].

Advanced sperm selection methods that have hitherto been less studied also met the inclusion criteria defined for the present systematic review: microfluidics sperm sorting, Zeta-potential, birefringence, and laser beam. Microfluidic sperm selection has arisen as a promising method, mimicking the microgeometry of the female reproductive tract and promoting sperm movement that is more similar to movement in a natural environment [68]. Most of the studies (60%) assessing the impact of microfluidics on ART outcomes found no differences compared to traditional sperm selection methods [54,56,57]. Considering only studies with a large sample size, one involving 116 infertile couples found that microfluidics led to greater ART outcomes compared to conventional methods, whereas another, committing 91 patients, observed no differences in clinical pregnancies [54,58]. In the light of the aforementioned information, larger randomized trials are required to evaluate the potential beneficial effect of this promising method. Concerning Zeta-potential, whereas one of the two studies that applied this technology reported a beneficial effect on ART outcomes in a population of 30 couples [60], the other, involving 54 couples, found no differences [59]. It is worth mentioning that while Kheirollahi-Kouhestani et al. [60] used unselected semen specimens, samples with at least one altered seminal parameter were used in the study of Duarte et al. [59], which could explain these different results. With regard to birefringence, this method relies on the use of Nomarski interference contrast to evaluate the refringence associated with the orientation of nucleoprotein filaments. The study of Gianaroli et al. [61] reported an improvement in the proportion of high-quality embryos on day 3 and in their ability to implant and progress beyond 16 weeks of gestation in a population of 231 couples. In the same way, an improvement in fertility and cleavage rates was observed when a laser beam was applied for the detection and selection of viable but immotile sperm in a population of 77 couples with complete asthenozoospermia [62]. This would suggest that this technique could improve ICSI outcomes in asthenozoospermic patients. However, and because only a very low number of studies are available, more research into the aforementioned methods is needed.

### 4.2. Advanced Sperm Selection Techniques Increase ART Outcomes through an Improvement of Sperm Quality/Functionality Variables

As a secondary outcome, this systematic review aimed to address whether advanced sperm selection techniques improve sperm quality/functionality parameters. Ten out of twelve studies found that advanced selection techniques improved both sperm quality/functionality parameters and ART outcomes. Among the four articles assessing the effect of IMSI on sperm quality and fertility, three reported that IMSI-selected sperm showed less DNA fragmentation [30,36]. Other studies investigating the effects of this method on sperm quality observed that the presence of vacuoles in sperm nuclei were related to impaired sperm quality, chromosomal aneuploidies, chromatin condensation defects and DNA damage [31,69,70].In the case of PICSI, only one of the included studies looking into ART outcomes evaluated the effects on sperm quality, finding a negative relationship between hyaluronan-bound sperm and the incidence of DNA fragmentation [44]. This concurred with other research demonstrating that hyaluronan-bound sperm are more likely to exhibit intact DNA [71]. The effects of MACS on sperm quality were assessed in two of the included studies (Romany et al., 2014; Merino-Ruiz et al., 2019). While no differences on sperm motility were found after MACS in the study with the largest population (263 samples) [49], another report, involving 92 samples, observed an improvement in sperm motility, viability, and morphology. Related to this, other research not included in this review—as it only evaluated the effects of MACS on sperm quality—found that sperm selected through this method exhibited lower DNA fragmentation, and higher mitochondrial membrane potential, sperm motility, and morphology [72,73]. Regarding microfluidics, only two of the included studies assessed sperm quality and found greater sperm motility and morphology and DNA integrity [55,58]. Again, these results agree with other reports showing similar outcomes [74,75]. As far as Zeta-potential is concerned, studies assessing sperm quality observed lower sperm DNA fragmentation after sperm selection, leading to an increase in ART outcomes [59,60]. In another study comparing Zeta-potential to other sperm selection methods, both MACS and Zeta-potential were able to increase the proportion of sperm with normal morphology and an intact DNA [76]. In addition, Zeta-potential was seen to be more efficient than binding to hyaluronic acid in the selection of sperm with an intact DNA [77]. Clinical evidence supports sperm DNA damage as a detrimental factor for reproductive outcomes [78,79]. Furthermore, the presence of sperm undergoing apoptotic-like changes in semen is related to male infertility [80,81]. In eight of the nine included studies, advanced selection methods were demonstrated to be better than the traditional ones to select sperm with an intact DNA [25,30,36,44,55,58,59,60]. Thus, only one study, involving 255 couples, found no differences in this parameter [29].

### 4.3. Strengths and Limitations

Overall, for the present systematic review, a great variability in the inclusion/exclusion factors was observed among the selected studies, highlighting the difficulty of comparing the results. All these factors underlie a potential risk of bias in systematic reviews, including the current one. For this reason, more studies including larger sample size and considering specific inclusion/exclusion factors are necessary to elucidate the effects on a specific cohort of infertile patients. In addition to that, publication bias might be present, as negative results may not be published to the same extent as positive outcomes do.

## 5. Conclusions

The comparison of advanced vs. traditional sperm selection methods (swim-up and density gradients) evidenced that the application of the former leads to an improvement in ART outcomes. Because the efficiency of such an improvement was found to be similar between methods, none appear to be better than another when dealing with the entire infertile population. This supports the need to define under which clinical conditions a particular method is more useful. Finally, this systematic review supports that not only do advanced selection methods improve ART outcomes, but also improve semen quality parameters, which leads one to suggest that the increase in the latter has a positive repercussion on the former.

## Figures and Tables

**Figure 1 ijms-23-13859-f001:**
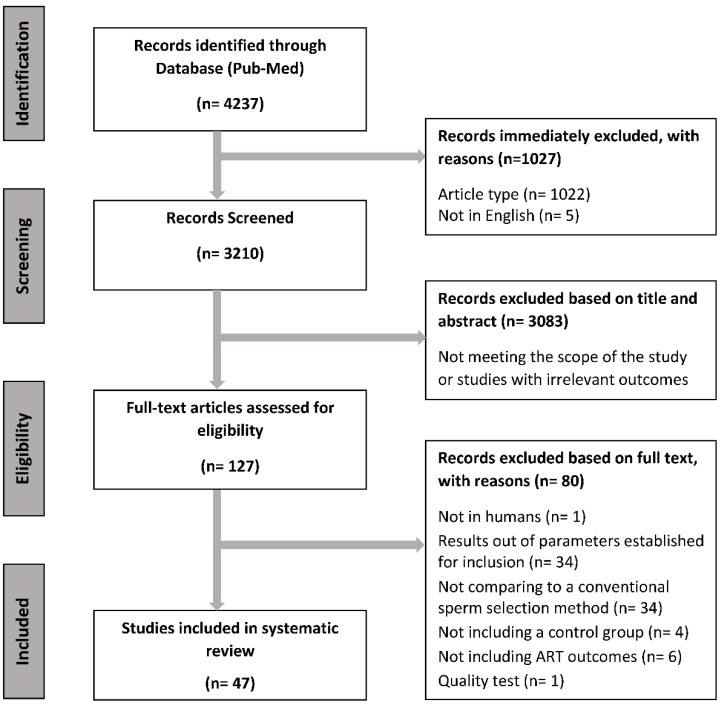
Flowchart of the literature search and selection procedure.

**Table 1 ijms-23-13859-t001:** Population, intervention, comparison, outcome, and study (PICOS) design, comprising inclusion and exclusion criteria and the keywords used for the definition of the search strategy and the eligibility of the study.

Parameter	Inclusion	Exclusion	Keywords
Population	- Infertile human males	- Studies conducted in species other than humans	Human, *Homo sapiens,* male, men, man, mammals, infertile, infertility, fertility
Intervention	- Primary: Studies assessing the effect of advanced sperm selection methods on assisted reproduction outcomes - Secondary: Studies assessing the putative influence of advanced sperm selection methods on sperm quality/functionality parameters	- Studies not assessing the effect of advanced sperm selection methods - Studies not including assisted reproductive outcomes or sperm quality/functionality parameters	Sperm selection, sperm quality, sperm function, sperm selection methods, sperm selection techniques,MSOME, motile sperm organelle morphology examination, birefringence, polarized light microscope, Raman microscopy, microfluidics, IMSI, intra-cytoplasmic morphologically selected sperm injection, hyaluronic acid, MACS, magnetic-activated cell sorting, swim up, density gradients, Zeta-potential, electrophoresisAI, artificial insemination, insemination, IVF, in vitro fertilization, ICSI, intracytoplasmic sperm injection, PICSI, physiological intra-cytoplasmic sperm injection.
Comparison	- For fertility outcomes: different advanced vs. conventional sperm selection method (e.g., density gradient or swim-up) - For sperm quality/functionality parameters: different advanced sperm selection methods vs. no sperm selection	- For fertility outcomes: Studies that did not compare advanced vs. conventional sperm selection methods on assisted reproduction outcomes - For sperm quality/function: studies that did not evaluate the effect of advanced sperm selection methods on sperm quality/functionality parameters	
Outcomes	- Assisted reproduction outcomes: Primary: Pregnancy rate, implantation rate, live birth rate Secondary: Fertilization rate, blastocyst rate, embryo quality - Sperm quality/functionality parameters: Motility Morphology DNA damagea		Sperm quality, morphology, oxidative, free radicals, ROS, oxidative stress, DNA damage, DNA fragmentation, oxidative damage, motility, viability, embryo, blastocyst, zygote, fertility, pregnancy, implantation, live birth, fertilization
Study design	- Research Article - Observational Study - Cross-sectional - Comparative - Longitudinal study	- Review article - Systematic reviews - Letters - Commentary articles - Case reports - Meta-analyses - Not written in English	Classical Article, Clinical Study, Clinical Trial, Clinical Trial, Phase I, Clinical Trial, Phase II, Clinical Trial, Phase III, Clinical Trial, Phase IV, Research study, Comparative Study, Corrected and Republished Article, English Abstract, Journal Article, Observational Study, English longitudinal study, cross-sectional study, Multicenter Study, Observational Study, Randomized Controlled Trial

**Table 2 ijms-23-13859-t002:** Relevant data of articles included in the systematic review. Abbreviations: ART: assisted reproduction technology; BMI: body mass index; FSH: follicle stimulating hormone; ICSI: intra-cytoplasmic sperm injection; IMSI: intra-cytoplasmic morphologically selected sperm injection; IVF: in vitro fertilization; MII: metaphase II; MACS: magnetic-activated cell sorting; MSOME: motile sperm organelle morphology examination; PICSI: physiological intra-cytoplasmic sperm injection; PVP: polyvinylpyrrolidone.

Reference	Aim	Advanced Sperm Selection Technique	Sample Size	Female/Male Inclusion/Exclusion Factors	Fertility Parameters Assessed	Main Results	Conclusions	Does It Improve Fertility Outcomes?
[16]	To compare IMSI vs. conventional ICSI in patients with severe oligoasthenoteratozoospermia.	IMSI	446 couples (IMSI: 227; ICSI: 219)	Inclusion: - Female age ≤ 35 years - Patients with severe oligoasthenoteratozoospermia	- Implantation rate - Pregnancy rate - Miscarriage rate	Patients with severe infertility subjected to IMSI showed significantly higher clinical pregnancy rates than when subjected to conventional ICSI.	IMSI leads to higher pregnancy rates compared to conventional ICSI in patients with severe oligoasthenoteratozoospermia.	Yes
[17]	To compare the clinical outcome of IMSI vs. conventional ICSI in unselected infertile couples.	IMSI	168 cycles (IMSI: 87; ICSI: 81)	NA	- Fertilization rate - Embryo quality - Implantation rate - Pregnancy rate - Live birth rate	Although IMSI did not improve overall clinical outcomes, a positive effect in implantation rates, clinical pregnancy, and live births was observed.	IMSI and conventional ICSI procedures provide similar clinical and laboratory results in an unselected infertile population.	No
[18]	To evaluate whether IMSI improves ICSI pregnancy rate in infertile couples with repeated failure.	IMSI	112 couples (IMSI: 62; ICSI: 50)	Inclusion: - Female age ˂ 37 years - ˃3 M-II oocytes retrieved - Two failures in ICSI cycles	- Fertilization rate - Embryo quality - Number of transferred embryos - Implantation rate - Pregnancy rate - Miscarriage rate	Fertilization rate, percentage of high-quality embryos, number of embryos transferred, and pregnancy rate were significantly higher in IMSI than in conventional ICSI.	Fertility outcomes are greater in IMSI than in conventional ICSI, in infertile couples with repeated failure.	Yes
[19]	To evaluate whether microinjection of sperm with a normal nuclear shape but large vacuoles affect pregnancy outcome	IMSI	56 couples (IMSI: 28; ICSI: 28)	Inclusion: - Female age < 40 years - >3 MII oocytes retrieved	- Fertilization rate - Embryo quality - Number of transferred embryos - Implantation rate - Pregnancy rate - Miscarriage rate	Microinjection of sperm with a normal nuclear shape but large vacuoles led to a significantly lower pregnancy rate per cycle and higher miscarriage rate per pregnancy compared to microinjection of sperm with normal nuclear shape.	Microinjection of sperm with vacuoles reduces pregnancy rates and is associated with early abortion.	No
[20]	To assess whether spermatozoa with strictly normal nucleus improves ICSI outcomes.	IMSI	160 couples (IMSI: 80 couples; ICSI: 80 couples)	NA	- Fertilization rate - Implantation rate - Pregnancy rate/per transfer - Miscarriage rate/per pregnancy	The percentage of high-quality embryos, implantation, and pregnancy rates were significantly higher and miscarriage rate lower in the IMSI group compared to the ICSI group.	IMSI improves reproductive outcomes.	Yes
[21]	To evaluate IMSI vs. conventional ICSI on ART outcomes.	IMSI	84 couples (IMSI: 51 couples; ICSI: 33 couples)	Exclusion: - Male age > 43 years - Females diagnosed with severe endometriosis	- Fertilization rate - Embryo kinetics - Embryo quality	Embryonic developmental parameters, clinical pregnancy rate, and the proportion of euploid embryos did not differ between IMSI and conventional ICSI groups.	IMSI does not improve embryo kinetics and quality, or clinical pregnancy rate compared to conventional ICSI.	No
[22]	To compare the reproductive outcomes of IMSI vs. IVF and ICSI.	IMSI	75 couples (Previous IVF failures: 22; Previous ICSI failures: 53)	Inclusion: - Females with normal hysteroscopy and/or 3D ultrasound scanning.	- Embryo quality - Blastocyst rate - Fertilization rate - Pregnancy rate - Live birth rate	Fertilization rates were significantly higher after IMSI than after IVF, but similar to ICSI. The percentage of high-quality embryos, the average number of blastocysts and transferred embryos were significantly higher after IMSI than after conventional IVF or ICSI.	IMSI leads to increase embryo quality and the number of transferred embryos compared to IVF or ICSI.	Yes
[23]	To compare IMSI vs. conventional ICSI in terms of neonatal outcomes.	IMSI	848 couples (IMSI: 275; ICSI: 573)	NA	- Fertilization rate - Blastocyst rate - Number of embryo transfers - Pregnancy rate - Live birth rate - Miscarriage rate	A significantly higher rate of multiple pregnancies was found in the IMSI compared to the ICSI group. A lower, but not statistically significant, proportion of congenital malformations was observed in the IMSI compared to the ICSI group.	IMSI could improve neonatal outcomes.	No
[24]	To define the indications for IMSI vs. conventional ICSI in infertile couples with two previous ICSI failures.	IMSI	216 couples (IMSI: 89; ICSI: 127)	Inclusion: - Two previous ICSI failures	- Fertilization rate - Pregnancy rate - Implantation rate - Total number of frozen embryos per cycle	Fertilization rate and the number of mature oocytes were significantly higher after IMSI than after ICSI. No differences were observed in the other fertility parameters.	In couples with two previous ICSI failures, IMSI does not improve clinical outcomes.	No
[25]	To assess the usefulness of IMSI in couples with repeated ICSI failure.	IMSI	125 couples	Inclusion: - Normal ovarian reserve - Absence of endometriosis - Female age < 38 years - Couples with repeated ICSI failures	- Fertilization rate - Cleavage rate - Embryo morphology - Pregnancy rate - Implantation rate - Live birth rate	IMSI resulted in significantly higher clinical pregnancy, clinical implantation, delivery, and birth rates compared to the last attempt of conventional ICSI in the same couples.	IMSI improves reproductive outcomes in couples with repeated ICSI failure.	Yes
[26]	To analyze the effect of IMSI in infertile couples for selecting patients who may benefit from this procedure.	IMSI	142 cycles (IMSI: 72; ICSI: 70)	Inclusion: - Female age < 37 years - Poor response (FSH) > 10 mIU/mL) - BMI > 25 Kg/m^2^	- Fertilization rate - Embryo quality - Pregnancy rate - Implantation rates	IMSI resulted in a significant increase in fertilization and high-quality embryo rates compared to ICSI in male factor infertility and in repeated implantation failure patients. No effect, however, was observed in the unselected group of patients.	The application of IMSI is beneficial for a selected group of patients with male factor infertility and repeated implantation failure.	Yes
[27]	To compare IMSI in infertile couples with male factor infertility and poor embryo development in their previous ICSI attempts.	IMSI	57 couples (IMSI: 20; ICSI: 37)	Exclusion: - Patients with endometriosis - Polycystic ovaries - Male age ˃ 42 years	- Blastocyst rate - Implantation rate - Pregnancy rate per cycle - Number of transfers - Rate of Arrested embryos - Pregnancy rate - Miscarriage rate	IMSI was better than ICSI in relation to the number of blastocysts/cycle, number of cycles with all embryos arrested and cycles without embryo transfer. No miscarriages were found in the IMSI group, but two out of three pregnancies in the ICSI group ended in miscarriage.	IMSI improves ART outcomes compared to ICSI, leading to a higher number of transferable embryos in infertile couples with male infertility and poor embryo development.	Yes
[28]	To compare clinical outcomes between IMSI vs. ICSI in patients with isolated teratozoospermia.	IMSI	122 cycles (IMSI: 52; ICSI: 70)	Inclusion: - >6 MII oocytes retrieved - Patients with isolated teratozoospermia Exclusion: - Female patients with endometriosis and polycystic ovaries	- Fertilization rate - Blastocyst rate - Arrested embryos - Pregnancy rate	Compared to ICSI, a significantly lower number of embryos arrested at early developmental stages, and a greater clinical pregnancy rate was observed when IMSI was applied.	IMSI improves ART outcomes in patients with isolated teratozoospermia.	Yes
[29]	To determine whether IMSI improves the semen characteristics and reproductive outcomes.	IMSI	255 couples, (IMSI: 116; ICSI: 139)	Exclusion: - Male age ˃ 39 years - Day 3: FSH level ˃ 9 UI/L	- Fertilization rate - Implantation rate - Pregnancy rate - Delivery rate	IMSI did not improve clinical outcomes (implantation, clinical pregnancy, and live birth rates) compared to ICSI.	IMSI has no benefit in the first ART attempt.	No
[30]	To compare DNA fragmentation, apoptosis and transcript levels in spermatozoa selected using IMSI, compared to conventional ICSI. In addition, embryo kinetics with time-lapse imaging and clinical outcomes were assessed and compared.	IMSI	80 couples (IMSI: 40; ICSI: 40)	Inclusion: - Healthy females without fertility problems - Female age ˂ 38 years - Basal FSH < 10 IU/mL - BMI between 25 and 30 Kg/m^2^ - A minimum of six mature oocytes. Exclusion: - Women with > 2500 pg/mL estradiol levels at the time of triggering - Cycles with no transferable embryos	- Biochemical pregnancy rate - Clinical pregnancy rate - Live birth rate - Implantation rate - Fertilization rate	Fertilization and embryo development were found to be improved after IMSI, in comparison to ICSI. The rates for implantation, biochemical and clinical pregnancy, and live birth were higher in IMSI group; however, only implantation rates showed statistically significant results.Cleavage abnormalities (fragmentation, multinucleation, uneven blastomere, and reverse cleavage) were less frequent in embryos derived from IMSI. Indeed, embryos derived from ICSI cleaved quickly and exhibited a greater number of abnormalities compared to IMSI.	IMSI improves sperm quality in terms of DNA damage and apoptosis. Additionally, IMSI leads to improved clinical outcomes and embryo kinetics in infertile patients.	Yes
[31]	To evaluate whether IMSI may influence embryo quality at day 2 compared to conventional ICSI.	IMSI	331 couples (IMSI: 159; ICSI: 172)	NA	- Fertilization rate - Embryo quality	No differences in terms of fertilization rate, early embryo cleavage rate, cleavage rate, and day 2 embryo quality were observed between ICSI vs. IMSI.	ICSI and IMSI show similar performance in terms of embryo quality on day 2.	No
[32]	To compare the reproductive outcomes of ICSI vs. IMSI in couples with implantation failure.	IMSI	200 couples (IMSI: 100; ICSI: 100)	Inclusion: - Normal karyotype - Implantation failure Exclusion: - Female age ˃ 39 years - Uterine defects - Infections - Endocrine problems - Coagulation defects - Hydrosalpinx	- Fertilization rate - Number of transferred embryos - Implantation rate - Pregnancy rate - Miscarriage rate - Ongoing pregnancy - Live birth rate	No statistically significant differences were observed between the two groups in all parameters assessed. However, miscarriage, ongoing pregnancy, and live birth rates showed better results (but not statistically significant) in the IMSI group.	IMSI does not improve clinical outcome in couples with implantation failure.	No
[33]	To examine whether IMSI improves ART outcomes in cases of advanced maternal age.	IMSI	66 cycles (IMSI: 33; ICSI: 33)	Inclusion: - Women with good health - Female age > 37 years - Normal basal FSH and LH levels - BMI < 30 kg/m^2^ Exclusion: - Polycystic ovaries - Endometriosis	- Fertilization rate - High-quality embryos rate - Blastocyst formation rate - Cycles with embryo transfer - Number of transferred embryos - Implantation rate - Pregnancy rate	Blastocyst formation rate, number of embryos transferred, implantation, and clinical pregnancy rates after IMSI were significantly higher than after ICSI.	IMSI in couples with advanced maternal age increases clinical pregnancy rates.	Yes
[34]	To analyze whether IMSI affects ART outcomes in couples with poor ovarian response.	IMSI	414 cycles (Normal responders, 324 cycles: 164 ICSI and 160 IMSI. Poor responders, 90 cycles: 43 ICSI and 47 IMSI)	Inclusion: - Normal responders: patients with more than 4 oocytes retrieved - Poor responders: Patients with less than 4 oocytes retrieved	- Fertilization rate - Embryo quality - Number of transferred embryos - Pregnancy rate - Implantation rate - Miscarriage rate	Normal responder group: no differences in terms of cycle outcomes were observed between ICSI- and IMSI-treated couples.Poor responder group: fertilization rate, proportion of cycles with embryo transfer, and number of embryos transferred were significantly lower in IMSI than in ICSI.	IMSI does not improve ART outcomes in couples with poor response to controlled ovarian stimulation.	No
[35]	To assess the potential beneficial effect of IMSI in couples with at least three repeated ICSI failure cycles.	IMSI	207 cycles (IMSI: 53; ICSI:154)	Inclusion: - Infertile women with at least three previous failure cycles of IVF–ICSI	- Fertilization rate - Cleavage rate - Embryo quality - Embryo kinetics. - Number of embryo transfers - Implantation rate - Biochemical pregnancy rate - Pregnancy rate - Live birth rate	Rates of implantation rate, clinical pregnancy and delivery were significantly higher in the IMSI than in the IVF–ICSI group. The proportion of miscarriages after IMSI was lower than after IVF of ICSI.	IMSI improves pregnancy outcomes in couples with more than three IVF–ICSI failures.	Yes
[36]	To assess whether MSOME improves ART outcomes compared to traditional ICSI.	IMSI	250 couples (IMSI: 125; ICSI: 125)	Inclusion: - Female menstrual cycle ranged 24–35 days Exclusion: - Female basal FSH was >10 IU/l - BMI > 29 kg/m^2^ - Polycystic ovarian syndrome - Endometriosis - Autoimmune, thyroid or chromosomal abnormalities	- Fertilization rate - Embryo quality - Number of transferred embryos - Number of embryo transfers - Implantation rate - Pregnancy rate - Live birth rate	Pregnancy and implantation rates were higher when IMSI was applied compared to ICSI. However, no difference in the proportion of pregnancies that led to a live birth was observed when IMSI and ICSI were compared.	MSOME ameliorates outcomes.	Yes
[37]	To investigate whether sperm selection by hyaluronic acid binding (PICSI) could improve ART outcomes in ICSI cycles.	PICSI	18 couples with 219 oocytes (HA-bound: 107 oocytes; Control: 112 oocytes)	Inclusion: - Female age range: 30 to 42 - Serum FSH level on menstrual day 3 ≤ 20 IU/L - ≥ 4 oocytes retrieved	- Fertilization rate - Blastocyst rate - Number of transferred embryos - Pregnancy rate - Implantation rate	After IMSI, fertilization and cleavage rates on day 2 in oocytes injected with hyaluronic acid-bound sperm were lower but not significantly different from after conventional ICSI. Blastocyst formation rate and the number of embryos transferred were similar between the two groups.	Sperm selection by hyaluronic acid binding does not improve ART outcomes after ICSI.	No
[38]	To compare fertility outcomes in conventional ICSI vs. sperm selection procedure based on hyaluronic acid binding ability in couples with male factor infertility.	PICSI	56 cycles (PICSI: 19; ICSI: 37)	Inclusion: - Patients with moderate or high male-infertility factor Exclusion: - Cycles with testicular sperm	- Fertilization rate - Cleavage rates - Biochemical pregnancy rate - Pregnancy rate - Miscarriage rate	Biochemical and clinical pregnancy rates were significantly higher in the PICSI group compared to the ICSI group. No differences in the abortion rate were found between groups.	PICSI increases pregnancy rates in couples with male factor infertility.	Yes
[39]	To evaluate the effect of sperm selection procedure based on hyaluronic acid binding ability on ART outcomes in couples with severe teratozoospermia.	PICSI	152 couples (PICSI: 77; ICSI: 75)	Inclusion: - Severe teratozoospermia	- Fertilization rate - Embryo quality - Pregnancy rate - Implantation rate - Miscarriage rate	Fertilization rate per retrieved oocyte, fertilization rate per inseminated oocyte, and the rate of high-quality embryos were significantly higher in the PICSI than in the ICSI group.	PICSI improves fertilization and embryo quality in couples with severe teratozoospermia.	Yes
[40]	To analyze ART outcomes comparing the PVP-ICSI and hyaluronic acid-ICSI (PICSI) sperm selection methods.	PICSI	21 couples with 206 oocytes (PICSI: 103; ICSI: 103)	Exclusion: - Low sperm quality (<10^6^ sperm/mL)	- Fertilization rate - Live birth rate - Embryo quality	A higher incidence of abnormal fertilization rate was observed in the PVP-ICSI group compared to the PICSI group. The proportion of high-quality embryos was similar in both groups.	PICSI improves fertilization rates.	Yes
[41]	To examine whether sperm selection by hyaluronic acid binding helps improve ICSI outcomes.	PICSI	156 couples (PICSI: 78; ICSI: 78)	Exclusion: - Female age > 38 years - Presence of uterine anomalies - Hydrosalpinx - Moderate or severe endometriosis - ˂3 oocytes retrieved	- Fertilization rate - Number of high-quality embryos - Implantation rate - Pregnancy rate - Live birth rate - Miscarriage rate	No differences in fertilization rate, number of high-quality embryos and clinical pregnancy rates were observed between ICSI and PICSI groups.	PICSI does not improve ART outcomes.	No
[42]	To investigate the efficiency of sperm selection based on hyaluronic acid binding ability vs. standard ICSI in terms of live birth rate.	PICSI	2766 couples (PICSI: 1386; ICSI: 1380)	Inclusion: - Female age range: 18 to 43 years - BMI between19–35 kg/m^2^ - FSH between 3–20 mIU/mL or an anti-müllerian hormone > 1.5 pmol/L	- Pregnancy rate - Miscarriage rate - Live birth rate	Miscarriage rates were significantly lower in the PICSI than in the ICSI group. Clinical pregnancy or preterm birth were different between the two groups.	PICSI does not improve clinical pregnancy rates but reduces miscarriage.	No
[43]	To examine the effect of sperm selection procedure based on hyaluronic acid binding ability on ART outcomes compared to conventional ICSI.	PICSI	250 couples (PICSI: 110; ICSI: 140)	Inclusion: - Female age ˂ 40 years - Regular (21–35 days) menstrual cycles - Normal baseline FSH level (12 IU/L)	- Fertilization rate - Implantation rate - Pregnancy rate - Live birth rate - Miscarriage rate	Fertilization, implantation, clinical pregnancy, and live birth rates were significantly higher and pregnancy loss significantly lower in PICSI compared to conventional ICSI.	Sperm selection based on hyaluronic acid binding is useful to improve clinical pregnancy rates.	Yes
[44]	To assess whether sperm selection based on hyaluronic acid binding ability affects ICSI outcomes	PICSI	50 couples (PICSI: 25; ICSI: 25)	Exclusion: - Polycystic ovarian syndrome - Endometriosis - Tubal factor - Repeated cycles	- Fertilization rate - Embryo quality - Number of embryos per transfer - Implantation rate - Pregnancy rate	Fertilization rate was significantly higher in the hyaluronic acid group compared to conventional ICSI. No differences, nevertheless, were found in the other fertility parameters assessed.	Sperm selection based on the ability to bind hyaluronic acid improves ICSI outcomes, in terms of fertilization rate.	Yes
[45]	To test whether the sperm selection procedure based on hyaluronic acid binding ability improves ART outcomes.	PICSI	379 couples (PICSI: 293; ICSI: 86)	Inclusion: - Female age ≤ 39 years	- Fertilization rate - Embryo quality - Implantation rate - Pregnancy rate - Miscarriage rate - Live birth rate	High-quality embryo and implantation rates were significantly higher in PICSI than in the control group. A trend towards a better pregnancy rate per transfer was also found in the PICSI compared to the control group.	Sperm selection based on the ability to bind hyaluronic acid is beneficial in ICSI treatments.	Yes
[46]	To evaluate whether the sperm selection procedure based on hyaluronic acid binding ability affects ICSI outcomes.	PICSI	680 couples (PICSI: 269; ICSI: 411)	Exclusion: - Female age > 40 years - <4 MII oocytes retrieved - Hyalurone binding < 2% - Testicular, donor, or cryopreserved sperm. - Patients undergoing preimplantation genetic diagnosis - Sperm count < 10^5^	- Fertilization rate - Implantation rate - Pregnancy rate - Miscarriage rate	Sperm selection procedure based on the ability to bind hyaluronic acid led to a significantly higher implantation rate and lower pregnancy loss compared to the control group.	Sperm selection through hyaluronic acid binding is beneficial for patients subjected to ICSI.	Yes
[47]	To evaluate clinical and embryo outcomes after sperm selection with MACS.	MACS	196 couples (MACS: 122; Control: 74)	Inclusion: - Maximum baseline FSH: 10 mIU/mL - Maximum baseline E_2_: 75 pg/mL - Female age < 35 years - No anatomical alterations - No history of low or absent ovarian response during FSH/HMG treatment	- Fertilization rate - Cleavage rate - Blastocyst rate - Implantation rate - Pregnancy rate	Cleavage and pregnancy rates were significantly higher in MACS group than in the control. Implantation rates did not increase after sperm selection by MACS.	MACS improves pregnancy rates in ICSI treatments.	Yes
[48]	To analyze the effectiveness of MACS in the removal of apoptotic sperm, in a population of patients subject to IVF/ICSI.	MACS	92 couples (MACS: 46; Control: 46)	NA	- Fertilization rate - Pregnancy rate	No differences in ART outcomes were found between MACS and the control. In spite of this, when couples were split into two groups based on the involvement of own or donated oocytes, MACS led to higher fertilization rates in the own oocyte group and greater clinical pregnancy in the donor oocyte group.	MACS improves fertilization and clinical pregnancy rates, in the case of donated oocytes.	Yes
[49]	To determine the impact of MACS on live-birth delivery rates after ICSI in couples with oocyte donation.	MACS	263 couples (MACS: 138; Control: 125)	Inclusion: - Female age range: 30 to 45 years - Body mass index < 30 kg/m^2^ - First ICSI cycle with oocyte donation - No uterine pathology - No history of miscarriage	- Fertilization rate - Embryo quality - Implantation rate - Pregnancy rate - Live-birth rate	Similar results were obtained for all fertility parameters assessed in the two groups.	Sperm selection by MACS technology does not improve the reproductive outcome of ICSI in couples undergoing oocyte donation.	No
[50]	To determine whether MACS increases live birth rate in couples presenting a high level of sperm DNA fragmentation.	MACS	305 couples (MACS: 87; Control: 218)	Inclusion: - Ejaculate volume > 1.5 mL - Sperm concentration > 5 × 10^6^/mL - Sperm progressive motility > 15% - Normal sperm morphology ≥ 1% - >5 M II oocytes retrieved Exclusion: - Poor ovarian response - Polycystic ovary syndrome - Adenomyosis - Endometriosis -Known genetic alteration - Uterine malformations	- Pregnancy rate - Miscarriage rate - Live birth rate	No differences in live birth rates were observed between MACS and the control. There was no evidence of miscarriage in the MACS group, whereas there were 10 miscarriages in the control.	Density gradient centrifugation followed by MACS in combination with ICSI has the potential to reduce the incidence of miscarriage in ICSI-derived pregnancies.	Yes
[51]	To evaluate whether the elimination of apoptotic sperm through MACS improves ICSI outcomes.	MACS	74 couples (MACS: 37; Control: 37)	Inclusion: - Couples with ≥2 years of idiopathic infertility - No obvious male and female infertility factors	- Fertilization rate - Total number of embryos - Embryo quality - Blastocyst rate - Number of transferred embryos - Pregnancy rate - Live-birth rate	Fertilization and blastocyst rates were increased in MACS compared to the control. No differences were observed in pregnancy rates.	MACS increases fertilization and blastocyst rates.	Yes
[52]	To evaluate the beneficial effect of MACS on ICSI in patients with teratozoospermia.	MACS	26 couples (Half of the mature oocytes were fertilized with conventional ICSI, and the second half after MACS)	Inclusion: - Teratozoospermia, Exclusion: - Female age ˃ 36 years - Not a normal ovarian response to controlled ovarian hyperstimulation	- Fertilization rate - Embryo quality - Percentage of blastocysts - Number of embryo transfers - Percentage of good quality blastocysts - Implantation rate - Live-birth rate	A significantly higher percentage of high-quality blastocysts was found in MACS compared to the control in women older than 30 years.	Sperm selection of non-apoptotic spermatozoa by MACS may be a useful method in couples with male infertility due to teratozoospermia and when the female is older than 30 years.	Yes
[53]	To assess the efficiency of sperm selection by MACS in a prospective randomized trial.	MACS	62 couples (MACS: 29; Control: 33)	Exclusion: - Females age ˃ 42 years - ˂6 MII oocytes retrieved - Oocytes with poor quality	- Fertilization rate - Embryo quality - Implantation rate - Miscarriage rate	High quality embryos, implantation, and pregnancy rates were higher in MACS than in the control. No differences were found for fertilization rates.	Sperm selection by MACS can improve clinical ICSI outcomes.	Yes
[54]	To evaluate whether a microfluidic device improves embryo and clinical outcomes in ICSI cycles.	Microfluidic sperm sorting	181 couples (Microfluidics: 91; Control: 90)	Inclusion: - Females with normal reproductive organs - Absence of poor ovarian reserve - Female age range: 20 to 40 years	- Fertilization rate - Pregnancy rate - Ongoing pregnancy	No significant differences in clinical pregnancy and ongoing pregnancy rates were reported between groups.	Microfluidic device does not enhance ART outcomes.	No
[55]	To evaluate whether a microfluidic sperm sorting device is useful to select sperm with high chromatin fragmentation.	Microfluidic sperm sorting	15 couples (Microfluidics: 4; Control: 11)	NA	- Fertilization rate - Implantation rate - Pregnancy rate	A higher clinical pregnancy rate was observed when the microfluidic device was used, leading patients with repeated ART failure and compromised sperm DNA integrity to achieve pregnancy.	Microfluidic device improves pregnancy rates and is particularly useful in patients with repeated ART failure and disrupted sperm DNA integrity.	Yes
[56]	To investigate the putative beneficial effect of microfluidic sperm sorting device on clinical outcomes.	Microfluidic sperm sorting	81 couples (Half of the embryos were produced after microfluidics and the other half served as a control)	Inclusion: - Female age < 42 years - ≥5 MII oocytes retrieved	- Fertilization rate - Blastocyst rate - Embryo quality - Pregnancy rate - Miscarriage rate	No differences in terms of clinical pregnancy, live birth, and miscarriage rates were found between groups.	Neither laboratory results nor clinical outcomes are improved by sperm selection through microfluidics.	No
[57]	To analyze the effect of microfluidics sperm selection on the results of ICSI cycles in patients with unexplained infertility.	Microfluidic sperm sorting	122 couples (Microfluidics: 61; Control: 61)	Inclusion: - Female age < 37 years	- Fertilization rate - Cleavage rate - Number of transferred embryos - Pregnancy rate - Live birth rate	The number of high-quality embryos was significantly greater in the microfluidics than in the control group. No differences between groups were found in the other fertility parameters.	Sperm selection through microfluidics prior to IVF does not alter fertilization, clinical pregnancy, or live birth rates in couples with unexplained infertility.	No
[58]	To compare the effect of conventional sperm selection method vs. microfluidics selection on ART outcomes.	Microfluidic sperm sorting	428 couples (Microfluidics: 116; Control: 312)	NA	- Fertilization rate - Pregnancy rate	In recurrent ART failure patients, fertilization rate was higher in the microfluidic group compared to the control. No differences in pregnancy rates were observed.	Microfluidics can improve fertilization rates in patients with repeated ART failure.	Yes
[59]	To evaluate whether Zeta-potential can be used to select sperm with intact DNA in non-normospermic patients, and to assess the impact of this selection on fertility parameters.	Zeta-potential	54 couples who used oocyte donors	Inclusion: - Non-normozoospermic semen analysis - ICSI treatment with oocyte donor	- Fertilization rate - Cleavage rate - Embryo quality - Blastocyst rate	No differences in embryo development parameters were found when sperm were selected by zeta-potential.	Zeta-potential reduces DNA fragmentation but does not improve laboratory outcomes.	No
[60]	To evaluate the efficacy of Zeta-potential to recover sperm with intact chromatin, and to assess whether this procedure improves ICSI outcomes.	Zeta-potential	30 couples (Half of the oocytes fertilized with sperm selected by zeta-potential and the second half served as a control)	Exclusion: - Endometriosis - Tubal adhesion	- Fertilization rate - Cleavage rate - Embryo quality - Pregnancy rate - Implantation rate	Fertilization rates were significantly higher when sperm were selected by Zeta-potential. Pregnancy and implantation rates in the Zeta-potential group did not differ from the control.	Selection of sperm through Zeta-potential may lead to higher fertilization rates but does not improve pregnancy or implantation rates.	Yes
[61]	To evaluate polarization microscopy as a method for sperm selection before ICSI.	Birefringence	231 couples (Birefringence method + ICSI: 112; Conventional ICSI: 119)	Exclusion: - Female with obesity or diabetes - Polycystic ovary	- Fertilization rate - Embryo quality - Number of embryos transferred - Pregnancy rate - Implantation rate - Miscarriage rate	The proportion of high-quality embryos on day 3 and their ability to implant and progress beyond 16 weeks of gestation were higher when sperm were selected by birefringence.	Not only is birefringence a diagnostic tool, but it is also an accurate and novel method for sperm selection that improves ART outcomes.	Yes
[62]	To evaluate the effectiveness of laser to detect those viable sperm among immotile sperm for their use in ICSI cycles.	Laser beam	77 couples (Laser method: 45; Control: 32)	Inclusion: - Patients with complete asthenozoospermia	- Fertilization rate - Number of transferred embryos	Fertilization and cleavage rates were significantly higher when sperm were selected by laser, compared to conventional sperm selection.	Application of a single laser shot for sperm selection improves fertilization and cleavage rates.	Yes

**Table 3 ijms-23-13859-t003:** Summary of all included research articles performing sperm quality analysis before and after the selection method. Abbreviations: IMSI: intra-cytoplasmic morphologically selected sperm injection; MACS: magnetic-activated cell sorting; MSOME: motile sperm organelle morphology examination; PICSI: physiological intra-cytoplasmic sperm injection.

Reference	Advanced Sperm Selection Technique	Primary Sperm Quality/Functionality Parameter Assessed	Main Results	Does It Improve Fertility Outcomes?
[29]	IMSI	- Sperm morphology - Chromatin protamination - Sperm DNA fragmentation	No differences for DNA fragmentation index, chromatin condensation and sperm morphology were found.	No
[30]	IMSI	- Sperm DNA fragmentation - Real Time PCR for evaluation of transcript levels related to apoptosis	A lower percentage of DNA fragmentation and transcript levels of apoptotic genes was observed in MSOME-selected spermatozoa.	Yes
[36]	IMSI	- Sperm DNA fragmentation	Incidence of DNA fragmentation was lower in sperm selected through IMSI.	Yes
[47]	MACS	- Sperm morphology	The percentage of sperm with normal morphology increased after selection by MACS.	Yes
[48]	MACS	- Sperm motility - Sperm viability - Sperm morphology	Sperm motility, viability and morphology were better after selection through MACS	Yes
[49]	MACS	- Sperm motility	Sperm selection by MACS did not alter the proportions of motile sperm.	No
[44]	PICSI	- Sperm DNA fragmentation - Chromatin condensation - Sperm morphology	A significant negative correlation between sperm bound to hyaluronic acid and DNA fragmentation, chromatin condensation, and sperm morphology was observed.	Yes
[60]	Zeta-potential	- Sperm DNA fragmentation - Chromatin protamination	The percentage of DNA-damaged sperm and chromatin condensation were significantly reduced when Zeta-potential was applied for sperm selection.	Yes
[55]	Microfluidic sperm sorting	- Sperm motility - Sperm morphology - Sperm DNA fragmentation	Sperm selected through a microfluidic device exhibited better motility and less DNA damage.	Yes
[58]	Microfluidic sperm sorting	- Sperm motility - Sperm morphology - Sperm DNA fragmentation	Microfluidics sperm sorting led to an increase in the percentage of sperm with low DNA fragmentation.	Yes
[62]	Laser beam	- HOS test	The percentage of sperm classified as viable by the HOS test was comparable to the percentage of sperm that exhibited a movement upon laser incidence.	Yes

## Data Availability

Data generated during the current study are available from the corresponding authors on reasonable request.

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
