# Peer review of "Advanced Sperm Selection Strategies as a Treatment for Infertile Couples: A Systematic Review"

_ijms, 2022, doi:10.3390/ijms232213859_

Round 1

Reviewer 1 Report

This review is well written and highlight the importance of ART in combating infertility. There is very little things that I am concerned about such as the first paragraph for the results section (Line 170-183). It feels like a repetition of Figure 1. Therefore, does that means figure 1 is both part of the results and materials and methods?

Table 2 should be adjusted for text to fit well to improve readability. This is probably editorial matter that may need the editor's attention since the journal may have specific format only for tables.

Author Response

Reviewer: This review is well written and highlight the importance of ART in combating infertility. There is very little things that I am concerned about such as the first paragraph for the results section (Line 170-183). It feels like a repetition of Figure 1. Therefore, does that means figure 1 is both part of the results and materials and methods?

Authors’ response: We would like to thank the reviewer for examining our manuscript providing constructive feedback, which helped us in the overall improvement of the quality of our manuscript. The reviewer is correct, because this figure shows in detail the process that has been carried out for the selection of articles as well as the number of articles included/excluded in each step.

Regarding the first paragraph of Results section, we agree that the information is similar. For that, and we have restructured the wording in order to look less repetitive to the reader.

Reviewer: Table 2 should be adjusted for text to fit well to improve readability. This is probably editorial matter that may need the editor's attention since the journal may have specific format only for tables.

Response: Thank you very much for your appreciation. We acknowledge that the table is long, but have tried to make it as understandable and readable as possible and as short as possible. Since now the manuscript is not at the final format for the journal, the publisher will adapt the format of this table to fit in the characteristics required. We remain open to the Editors and Reviewers to conduct any specific change that may help in the readability of this table.

Reviewer 2 Report

A brief summary

This paper covers a wide area of sperm selection strategies. This information is valuable for medical doctors and embryologists.

Broad comments

This manuscript is well-written and suitable for publication after minor revision.

Specific comments

Minor concerns

Line 15: B >> B OR Biotechnology of Animal and Human Reproduction (Techno Sperm) >> Biotechnology of Animal and Human Reproduction (Techno Sperm)

Line 23: Assisted >> Assisted

Line 382: D >> D

Line 390: A >> A

Line 392: S >> S

Line 425: 2022;54;. >> 2022;54:e14405.

Line 491: 2011;9. >> 2011;9:123.

Line 499: 846 >> 846.

Line 502: 2009; 339:. >> 2009;339:b2700.

Line 536: 2011;9. >> 2011;9:99.

Line 553: 1393 >> 1393.

Line 570: 2020;2020;. >> 2020;2(2):CD010167.

Line 601: 2019;51:. >> 51(10):e13403.

Author Response

Reviewer: A brief summary

This paper covers a wide area of sperm selection strategies. This information is valuable for medical doctors and embryologists.

Broad comments

This manuscript is well-written and suitable for publication after minor revision.

Response: We would like to thank the reviewer for their positive feedback on our manuscript and for examining thoroughly our manuscript providing useful feedback, which helped us in the improvement of the quality of our work. All the changes suggested by the reviewer have been made.

Reviewer: Specific comments

Minor concerns

Line 15: B >> B OR Biotechnology of Animal and Human Reproduction (Techno Sperm) >> Biotechnology of Animal and Human Reproduction (Techno Sperm)

Line 23: Assisted >> Assisted

Line 382: D >> D

Line 390: A >> A

Line 392: S >> S

Response:  All the changes suggested above were made according to the Reviewers’ suggestion.

Line 425: 2022;54;. >> 2022;54:e14405.

Line 491: 2011;9. >> 2011;9:123.

Line 499: 846 >> 846.

Line 502: 2009; 339:. >> 2009;339:b2700.

Line 536: 2011;9. >> 2011;9:99.

Line 553: 1393 >> 1393.

Line 570: 2020;2020;. >> 2020;2(2):CD010167.

Line 601: 2019;51:. >> 51(10):e13403.

Response:  All the changes suggested for the reference list were made according to the Reviewers’ suggestion.